# Preparation of Intumescent Fire Protective Coating for Fire Rated Timber Door

**Jessica Jong Kwang Yin [1], Ming Chian Yew [1,*] , Ming Kun Yew [2] and Lip Huat Saw [1]**

[1]  Department of Mechanical and Material Engineering, Lee Kong Chian Faculty of Engineering and Science, University of Tunku Abdul Rahman, Cheras, Kajang 43000, Malaysia; jessicajong727@gmail.com (J.J.K.Y.); sawlh@utar.edu.my (L.H.S.)

[2]  Department of Civil Engineering, Lee Kong Chian Faculty of Engineering Science, University of Tunku Abdul Rahman, Cheras, Kajang 43000, Malaysia; yewmk@utar.edu.my

*  Correspondence: yewmc@utar.edu.my or yewmingchian@gmail.com; Tel.: +6-039-086-0288

**Abstract:** Intumescent flame-retardant coating (IFRC) provides a protective barrier to heat and mass transfer for the most efficient utilization of a wide variety of passive fire protection systems at the recent development. This article highlights the fire-resistance, physical, chemical, mechanical, and thermal properties of the IFRC using a Bunsen burner, furnace, Scanning Electron Microscope, freeze-thaw stability test, Instron Micro Tester, and thermogravimetric analysis (TGA) test. The five IFRC formulations were mixed with vermiculite and perlite for the fabrication of fire-resistant timber door prototypes in this research project. Additionally, the best fire-resistance performance of the fire-rated door prototype was selected and compared with a commercial prototype under the fire endurance test. An inventive fire-rated door prototype (P2), with a low density of 636.45 kg/m$^3$, showed an outstanding fire-resistance rating performance, resulting in temperature reduction by up to 54.9 °C, as compared with that of the commercial prototype. Significantly, a novel fire-rated timber door prototype with the addition of formulating intumescent coating has proven to be efficient in preventing fires and maintaining its integrity by surviving a fire resistance period of 2 h.

**Keywords:** fire protection system; filler; fire rated timber door; flame retardant additives; intumescent coating

## 1. Introduction

There are two kinds of fire protection systems in buildings, active and passive fire protection systems, which are an integral part of any modern-day building to protect lives and assets by enhancing the fire prevention and protection techniques [1]. The active fire protection system includes a fire or smoke alarm, sprinkler, and fire extinguishers. However, this system has required some actions to work efficiently in the event of a fire and this might fail due to lack of maintenance or other possible problems such as frozen pipes and inadequate water pressure. Whereas passive fire protection (PFP) system includes fire or smoke dampers, fire doors, and firewalls. This PFP system plays an essential role to provide fire safety protection by using flame-retardant materials. Therefore, with regards to this research, a new and innovative fire-resistant timber door is incorporated with the intumescent flame-retardant coating (IFRC) that acts as an effective PFP system and its potential application for lightweight fire-rated timber door. The intumescent coating reacts under the influence of fire and swells in a controlled manner to many times its original thickness, and thus produces an insulation carbonaceous char or foam that protects the substrate from the effects of the fire [2]. By such effectiveness, intumescent coating plays a critical role in protecting the building, and it may also gain



extra time for building occupants to escape safely during the outbreak of fire by trapping the fire and smoke as well as insulating the heat.

IFRCs can either be coated as a thin film or act as a binder of PFPs to enhance the fire-resistance performance. Boards are rigid prefabricated materials that usually come along with hydraulic binders [3]. Common fire doors are made of magnesium oxide boards, gypsum boards and even heavy timber wood [4–6]. However, there is one common problem in all the existing commercial fire doors which is their high density [7]. According to Fire Industry Association, due to the sheer weight and force produced when the heavy fire doors are opened and closed (density > 1200 kg/m$^3$) [8], it may cause human injury especially in places such as residential care or nursing homes.

In this research study, intumescent coatings and lightweight flame-retardant materials such as vermiculite and perlite are used to construct the lightweight fire-resistant timber door prototypes (density is about 600 ± 50 kg/m$^3$) for 2-h fire rating. Intumescent paint is the recent trend in fire retarding products in construction building materials because of its many beneficial properties such as providing low-odor, lightweight, and is environmentally friendly. Exhaustive investigations have revealed that the intumescent flame retardant coating has achieved good flammability and physical and chemical performances though many linger widely on steel structure applications [9–12]. Up to today, the performance of fire-resistant boards has yet to be tested and investigated with the addition of intumescent fire protective coating for the development of a fire-resistant timber door. Furthermore, this research project has also highlighted huge potential for incorporating the intumescent coating composites, which consists of three main flame-retardant additives, namely ammonium polyphosphate (APP), an acid source; pentaerythritol (PER), a carbon source; and melamine (MEL), blowing agents, into the fire door. These are mixed in a weight ratio of 2:1:1 and are bonded together with flame retardant fillers and vinyl acetate (VA) copolymer as well as vermiculite and perlite which react together to form a protective thermal barrier at high temperature in fire doors [13]. It is important to note that flame-retardant additives are useful chemical compounds for fire retardant to provide varying degrees of flammability protection. A research study conducted by Xia and his co-workers has stated that adding too much APP into the intumescent formulation may reduce the mechanical properties of the intumescent coating. Hence, an appropriate ratio of APP/PER/MEL should be set prior to the development of intumescent coating [14]. One of the ingredients, titanium dioxide (TiO$_2$), is important to formulate an intumescent coating in which it does not only provide the usual properties of colour and opacity, but it also likely takes place in the intumescence process which stabilizes the insulating foam at high temperatures when most of the carbon oxidizes and burn off [15]. Besides that, polymer binder is also one of the final key materials to produce a uniform cellular structure to provide good thermal insulation. In a research study conducted by Wang and Yang, the influence of the binder, vinyl acetate (VA), copolymer emulsion in the water-based coating is used to minimize the smoke and toxic fume emission without compromising the quality and effectiveness of the intumescent coating in fire protection [16].

In addition to that, vermiculite and perlite act as flame-retardant materials promoting a very low density, high porosity, good thermal insulation properties, chemical inertness, and good fire-retardant materials. This makes them attractive to be used widely as pore-forming additives for heat insulation applications [17]. Moreover, the uniqueness of the materials can insulate the heat from the fire by expanding with small particles under high temperatures. Besides that, a highly porous aggregate that can absorb moisture in varying degrees, the presence of moisture in the aggregate would turn into steam and evaporate from the materials during a fire test, and this will extend the fire duration [18]. As mentioned, the performance of the IFRC is based on the appropriate materials combination and the compatibility of the flame-retardant fillers with a polymer binder. Therefore, it is essential that the flame-retardant fillers provide good fire protective performance and fire retarding efficiency properties [19,20]. The grouping of aluminium hydroxide, Al(OH)$_3$; calcium silicate, CaSiO$_3$; magnesium hydroxide, Mg(OH)$_2$; chicken eggshell (CES) powder (bio-filler derived from CES waste); and calcium carbonate, CaCO$_3$ is used in this research project [20]. This project aims to formulate an

appropriate combination of intumescent flame-retardant coating for the development of fire-resistant timber door prototype. The performances of fire protection, mechanical, chemical and physical properties of intumescent flame-retardant coatings are evaluated and investigated. This research work is intended to design, fabricate, and examine the fire-resistant timber door prototype by incorporating intumescent fire-protective coating in order to fulfil the 2-h fire rating.

## 2. Experimental Program

### 2.1. Materials and Sample Preparation

The first part of this project is to synthesize the water-based intumescent paint using a high-speed disperse mixer for 30 min at room temperature until it is completely homogenous. The formulations of intumescent coatings are tabulated in Table 1. APP, PER, and MEL were used as flame-retardant additives. The aluminium hydroxide, magnesium hydroxide, calcium silicate, chicken eggshell (CES) powder, calcium carbonate, and titanium dioxide were used as flame-retardant fillers, and vinyl acetate (VA) copolymer emulsion acts as a water-based polymer binder in this study. The fire resistance and physical properties of the intumescent coatings were characterized and assessed through a series of tests such as Bunsen burner test, furnace test, scanning electron microscope (SEM) analysis test, adhesion strength test, freeze-thaw cycle test, and thermogravimetric analysis (TGA) test.

**Table 1.** Specifications of experimental materials.

| Samples | Ingredients (wt.%) | | | | | | | | | |
| | Flame Retardant Additives | | | | Flame-Retardant Filler (Pigment) | Flame-Retardant Fillers | | | | |
| | APP | PER | MEL | Vinyl Acetate (VA) Copolymer Emulsion | $TiO_2$ | $Al(OH)_3$ | $Mg(OH)_2$ | $CaSiO_3$ | CES | $CaCO_3$ |
|---|---|---|---|---|---|---|---|---|---|---|
| J1 | 20 | 10 | 10 | 50 | 4.0 | 3.0 | – | 3.0 | – | – |
| J2 | 20 | 10 | 10 | 50 | 4.0 | 3.0 | – | – | 3.0 | – |
| J3 | 20 | 10 | 10 | 50 | 4.0 | – | 3.0 | 3.0 | – | – |
| J4 | 20 | 10 | 10 | 50 | 4.0 | – | 3.0 | – | 3.0 | – |
| J5 | 20 | 10 | 10 | 50 | 4.0 | – | 3.0 | – | – | 3.0 |

The second part of this project is to design and fabricate the fire-resistant timber door prototype with a dimension of 300 mm (length) × 300 mm (width) × 40 mm (thickness). The fire protective performance of the fire timber door prototype is determined through a small-scale fire test. Table 2 shows the details of the experimental prototypes.

**Table 2.** Details of experimental prototypes.

| Fire Door Prototype Dimensions: 300 mm (*L*) × 300 mm (*W*) × 40 mm (*H*) | P1 | P2 | P3 | P4 | P5 |
|---|---|---|---|---|---|
| Weight (g) | 2565 | 2578 | 2589 | 2605 | 2610 |
| Density (kg/m$^3$) | 633.44 | 636.45 | 639.23 | 643.22 | 644.52 |

Moreover, the fire endurance test and the temperature rise test are conducted to compare and determine the best fire timber door prototype as compared to the commercial prototype in terms of heat transmission and integrity failure or a significant leakage. The flow chart of the experimental work details and the engineering drawing of the fire-resistant timber door prototype as well as thermocouple locations (T1 and T2) are shown in Figure 1.

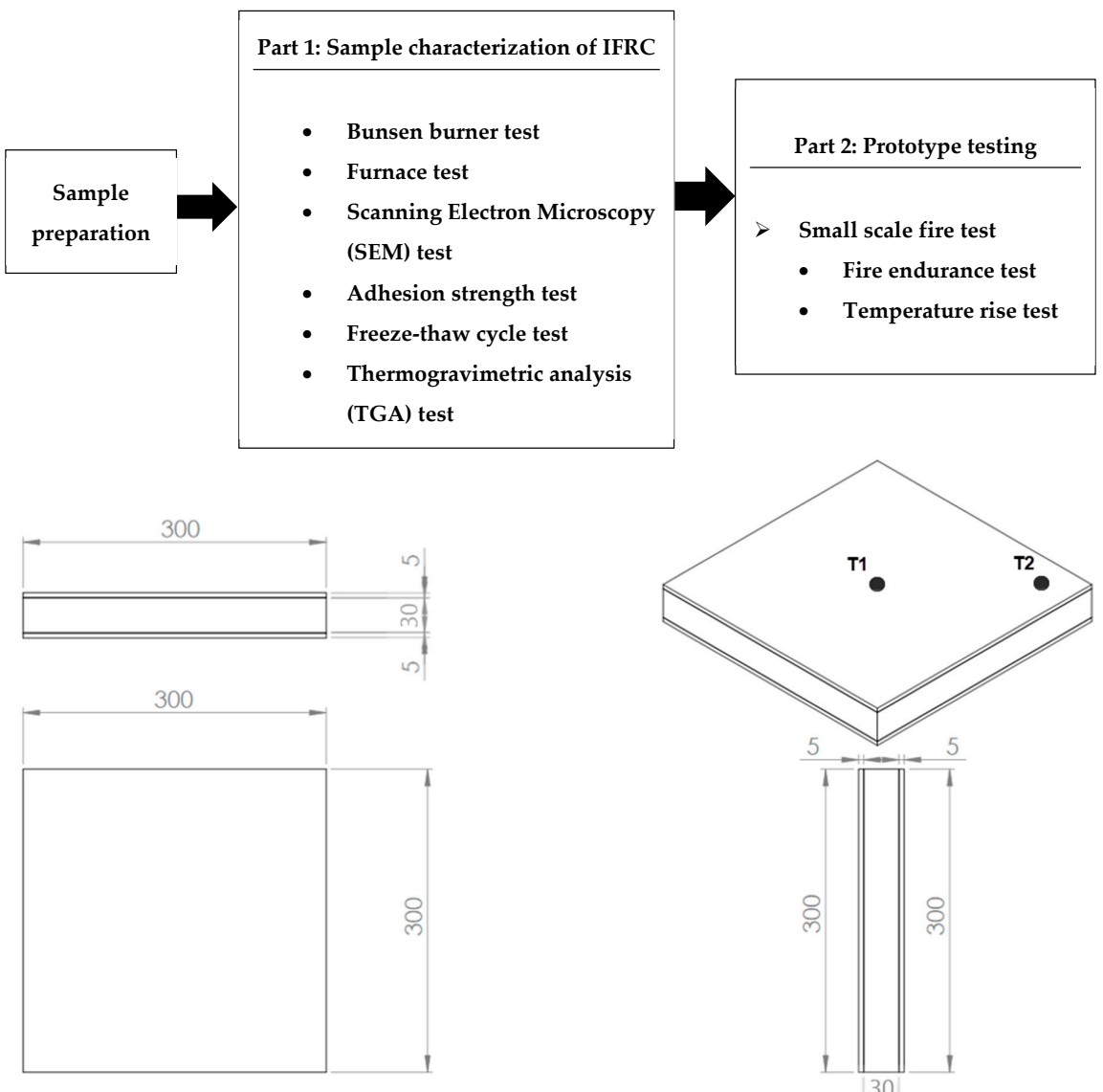

**Figure 1.** Flow chart of the experimental work and the dimensions of the prototype.

*2.2. Testing for Coated Samples*

2.2.1. Bunsen Burner Test

The coating samples were coated on a 100 mm × 100 mm × 3 mm steel plate with a thickness of 2 ± 0.2 mm. The thermocouple plate is attached at the backside of the steel plate coated with intumescent flame-retardant coating and then connected to a digital handheld thermometer for temperature measurement. Figure 2 shows the experimental setup of the Bunsen burner test for the coated samples. The sample was exposed to a Bunsen burner flame spray gun blow torch for 1 h at a temperature flame of about 1000 °C. The Bunsen burner gas consumption is about 160 g/h and the distance between the sample and Bunsen burner nozzle is about 7 cm.

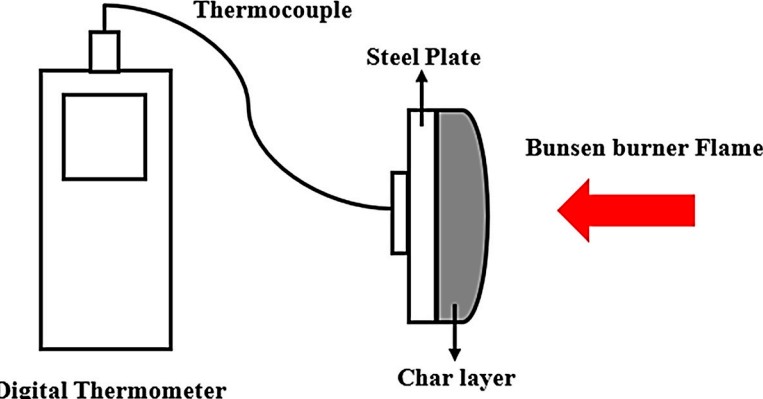

**Figure 2.** Schematic of the experimental setup for coated samples.

### 2.2.2. Furnace Test

For the furnace test, all steel plates (dimensions: 50 mm × 50 mm × 1 mm) were coated with IFRCs with a thickness of 2 ± 0.2 mm. The coated samples were then placed into the furnace (Barnstead Thermolyne 62700, Barnstead Thermolyne, Dubuque, IA, USA) with high temperatures of 500 and 600 °C, respectively (heating rate is 20 °C/min). After that, the thickness of the char layer for each sample was measured. Up to date, there is no standard test to study the strength of a char sample. In this experimental work, a small-scale compressive strength of char test was designed and conducted by stacking up the 50 g coin-shape metals one by one, which acts as a static load on the char layer for measuring and evaluating the char strength of the sample.

### 2.2.3. Scanning Electron Microscopy (SEM) Analysis

For SEM examination/analysis, all samples are initially coated with gold in order to eliminate the possible charging effects. A low beam energy with 1 kV is conducted to decrease the possibility of any thermal damage to the samples. The main objectives of the SEM analysis are to conduct a detailed examination using Hitachi EDAX S3400–N Scanning Electron Microscopy (SEM) machine manufactured by Hitachi High Technologies America, Inc. in America and analysis on the IFRC coating samples' morphology and thus to determine and identify the interfacial bonding of each type of coating sample on the steel substrate.

### 2.2.4. Adhesion Strength Test

Figure 3 shows the experimental setup for the measurement of adhesion strength of the coating samples. The samples were coated with a thickness of 2 ± 0.2 mm IFRC on one side of the cylindrical steel rod. After that, the coated cylindrical steel rod was then attached to a bare cylindrical steel rod using an epoxy glue with a thickness of 2 ± 0.2 mm. Two cylindrical steel rods were then tested using the Instron Micro Tester–Universal Testing machine manufactured by Instron in Norwood, MA, USA with a constant rate of 1 mm/min continually drawn apart in tensile mode until the coating samples on the cylindrical steel rods tore apart. The adhesion strength of the coating sample in MPa is calculated based on the following Equation (1) [21]:

$$f_\mathrm{b} = \frac{F}{A} \tag{1}$$

where, $f_\mathrm{b}$ = adhesion strength (MPa); $F$ = maximum load at rupture(N); $A$ = adhered area $\left(\mathrm{mm}^2\right)$.

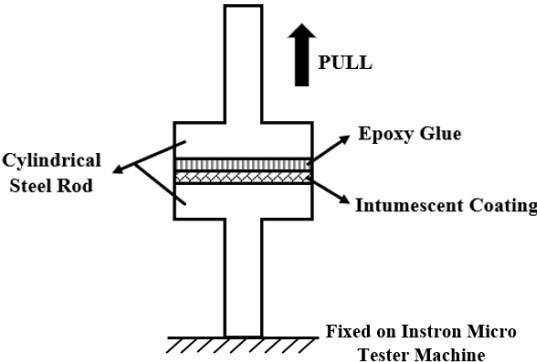

**Figure 3.** Schematic of the experimental setup for adhesion strength.

### 2.2.5. Freeze-Thaw Cycle Test

The freeze–thaw resistance test was used to determine the ability of the coating sample to withstand the highly destructive forces of cyclic freezing and thawing. The coating samples were coated onto different sets of 50 mm × 50 mm × 1 mm steel plates with a thickness of 2 ± 0.2 mm. The coating samples were placed in an air flow at 25 °C for 8 h and then placed in the freezer at −20 °C for another 8 h. Lastly, the coating samples were heated in a drying oven at 50 °C for another 8 h. This process is noted as a freeze–thaw cycle period [22].

### 2.2.6. Thermogravimetric Analysis (TGA) Test

Thermogravimetric analysis (TGA) was carried out using Perkin Elmer STA8000 model manufactured by Perkin Elmer Inc., Waltham, MA, USA. Test samples with about 5–8 mg thick thin films were placed in a crucible and heated from 30 to 1000 °C at a heating rate of 20 °C/min under nitrogen gas.

### 2.2.7. Testing for Fire Resistant Timber Door Prototype

● Small Scale Fire Test

The small-scale fire test was conducted on prototype 1–5; the intumescent binders for each prototype, which is based on the formulation J1–J5 that is provided in Table 1, is shown in Section 2.1. After that, the fire protection performance of the prototypes was tested using the Bunsen burner at about 1000 °C for 2 h. The temperature profile of the backside of the fire-resistant timber door prototype was measured and recorded using a digital handheld thermometer. Figure 4 shows a small-scale fire test experimental setup for the fire-resistant timber door prototype.

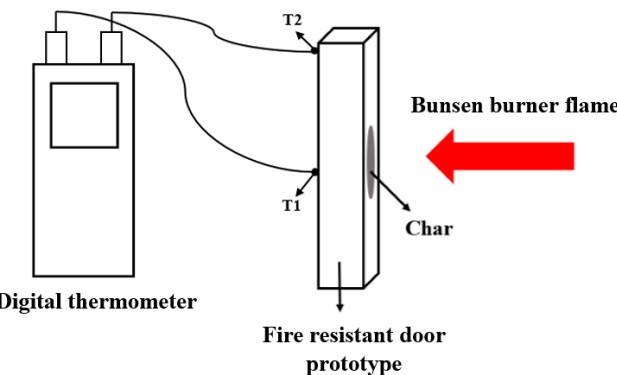

**Figure 4.** Schematic of the experimental setup for fire-resistant timber door prototype.

- Fire Endurance and Temperature Rise Tests for the Timber Door Prototype

The fire endurance test was carried out using the Bunsen burner to determine and compare the fire-resistance rating and integrity of the best prototype and commercial prototype. For the fire endurance test, the tested prototype can be qualified as a standard fire-rated timber door if it can maintain its integrity when exposed to 2 h fire (at about 1000 °C) without an integrity failure or displaying a significant leakage [23]. Whereas, the temperature rise is also recorded over a 30-min interval until 120-min, this is to examine the heat rise on the backside of the fire-resistant timber door prototype. The thermocouples are fixed at two points of the fire-resistant timber door prototype (centre (T1) and edge (T2)) to measure the transmission of heat from one point to another point of the prototype. This is to ensure that the prototype has taken adequate fire protection by reducing the heat transmission rate and flame propagation so that it can prolong the evacuation time for the building occupants if there is a fire.

## 3. Experimental Results and Discussion on Findings

### 3.1. Bunsen Burner Test

The purpose of this specified test is to characterize the physical and chemical reactions of the char formation of the IFRC. The temperature profiles of the steel plates coated with five IFRC formulations (J1–J5) and the bare steel plate were compared after 60 min of fire testing. In this experimental work, the temperature of 400 °C was chosen as the critical temperature for the coated sample with IFRC [24]. The fire protection results of the coated steel plates were then plotted as a function of time and presented in Figure 5.

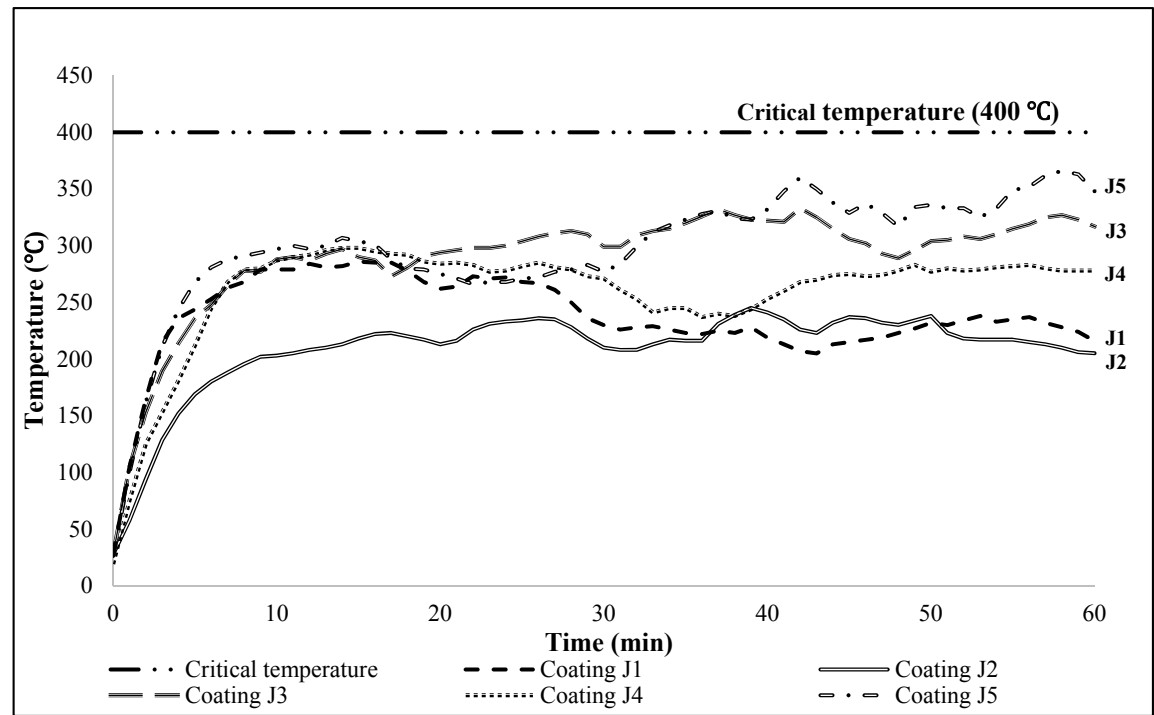

**Figure 5.** Evolution of temperature on the coated steel plates.

The thickness of the char layer and equilibrium temperature of the intumescent coatings with the influences of different types of flame-retardant fillers are shown in Table 3. The first 10 min have showed that there is a similar pattern of temperatures for the coated samples, which is below 300 °C. After that, the temperatures started to fluctuate at 20 min due to the physical and chemical reactions of the coating formulations during the high-temperature test. The temperature continues to rise for

30 min, it is observed that the coating J5 starts to rapidly increase until reaching 360 °C at 42 min. This may possibly be due to the fact this coated sample has started to lose its adherence strength or interfacial bonding from the steel plate during the fire test. Whereas coating J1 drops rapidly after reaching 205 °C at 43 min, this might due to the reduction of gas pressure from the Bunsen burner flame spray gun blow torch.

**Table 3.** Thickness of char layer and the equilibrium temperature of coated samples.

| Coating Samples | Thickness of Char Layer (±0.1 mm) | Equilibrium Temperature of Coating (°C) |
|---|---|---|
| Original Steel Plate | – | >Critical Temperature (400 °C) |
| J1 | 5.8 | 238 |
| J2 | 6.0 | 217 |
| J3 | 2.8 | 306 |
| J4 | 3.8 | 279 |
| J5 | 4.3 | 325 |

It is also observed that the temperature of all the coatings starts to fluctuate again until reaching 50 min. After 50 min of the test, the temperature reaches an equilibrium value and almost remains unchanged until the last stage of the test. The equilibrium temperature of coating J2 that is 217 °C is considered to be significantly lower than the other coatings and thus it has showed to have the best result for fire protective performance. The char combined the positive effects of the flame retardant additives, intumescent paint and fillers (3 wt.% of $Al(OH)_3$ and CES), promoting a good fire-protective barrier and producing the thickest and most excellent char layer (i.e. about 6.0 mm) to block the fire while safeguarding the formation of a uniform foam structure, which is in good agreement with other researchers, as highlighted by Wang and Yang, 2010.

With regards to the other observation, coating J5 has the highest equilibrium temperature of 325 °C, almost reaching the critical temperature (400 °C) after 40 min of fire exposure. The equilibrium temperature of coating J5 might be due to its char layer showing porosity and lightness, resulting in the loss of adherence of the char to the steel plate or by a loss of cohesion of the char layer; similar observations have been made by other researchers too [24].

*3.2. Furnace Test*

The purpose of the furnace test is to investigate and critically assess the fire protection performance of the coated samples by measuring the thickness of the char layer at temperatures of 500 and 600 °C. The thicknesses of the char layer have been measured and recorded in Figure 6. Based on the results, all of the char layers have expanded from 500 to 600 °C except coating J3. From the result obtained, it can be observed that the char layer of coatings J3, J4 and J5 with the addition of magnesium hydroxide has a slightly less thick char layer as compared to coatings J1 and J2. These phenomena are due to the flame-retardant filler (i.e., magnesium hydroxide) that has a negative contribution toward a fire-protective performance. It has a poor char expansion and resulted in failing to provide the required thick insulating barriers to protect the underlying steel plate from the heat. The thickness of coating J3 increased at 500 °C, this is because of the gaseous water phase during the decomposition of $Mg(OH)_2$ filler envelop the flame, and thereby excluding the oxygen and diluting flammable gases [25]. Whereas the thickness of the char layer of coating J3 decreased at 600 °C, this is due to the fire that propagated through the fire-protective barrier which thus led to a decrease of thickness of the char layer.

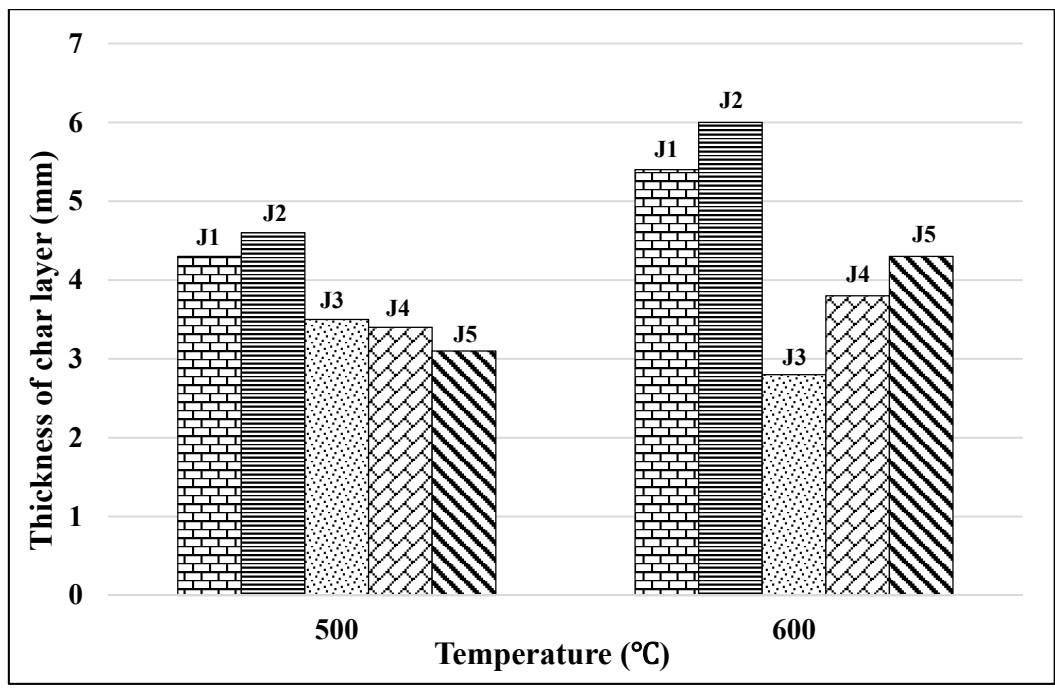

**Figure 6.** Thickness of char layer of the coated samples at 500 and 600 °C.

The coating J2 has the thickest char layer of 4.6 mm and 6.0 mm at temperatures of 500 °C and 600 °C, respectively. This indicates that it has the highest expansion value of a char layer due to the addition of $Al(OH)_3$ and CES that have significantly exhibited the expansion of the char layer. The formation of a cohesive structure during burning can potentially be initiated by decarbonation. Decarbonation of CES can release carbon dioxide which can trap the degraded product into the residue and induce swelling [11]. The thickness of the char layer of coating J1 was 4.3 mm at 500 °C and 5.4 mm at 600 °C. The expansion value of the thickness of the char layer of coating J1 is approximately 1.1 mm. The char layer of coating sample J1 is considered to be thicker than the coating samples J3, J4 and J5, which is again due to the addition of $Al(OH)_3$, providing the strong reversibility of dehydration reaction with water released inside the particle recombining to the reactive surface of the freshly formed alumina, resulting in a good flammability resistance filler and expansion of the char layer [25]. The expansion values of the thickness of the char layers of coatings J4 and J5 were 0.4 mm and 1.2 mm, respectively. Although the addition of CES in coating J4 and the addition of $CaCO_3$ in coating J5 showed significantly good flame-retardant fillers, the combination of CES and/or $CaCO_3$ with $Mg(OH)_2$ filler may have led to poorer fire-resistance performance. Hence, it can be deduced that $Mg(OH)_2$ filler can restrict the formation of the char layer by declining the fire-protection performance.

In addition, the strength of the char layer has also been determined for each coating sample after reaching the maximum heating temperature of 600 °C through the furnace test. The weight load of 50 g coin-shape metal is stacked up on-top of the char layer one by one until the char layer has completely cracked. In summary, only coatings J1, J2 and J5 have undergone this test as the char layers of coatings J3 and J4 already cracked after heating up to 600 °C in the furnace. The strength of the char layer of coating J2 can withstand the maximum load up to 650 g, as compared to coatings J1 (400 g) and J5 (600 g). This experimental test has proven that coating J2 can achieve the best fire protection performance and the strongest char layer against the fire.

### 3.3. Surface Morphology of Char Layer of Coated Samples

Microscopic analyses are executed with the scanning electron microscope (SEM) to examine and analyze the surface morphology of the intumescent coating. The SEM micrographic images of all char layers of coated samples after the Bunsen burner test are shown in Figure 7. The orange color indicates

the holes and voids in the char layer of the coated samples. The images of coatings J1, J3, J4 and J5 illustrated larger holes and voids. These holes are formed because of trapped gas by its blowing agent when the coating is exposed to fire [26]. The tiny holes and void structures can be the consequence of fire and heat propagation. The fire and heat propagated to the crack lines on the foam structure can also lead to a poor fire-protective performance of the coating [14]. On the other hand, the char layer of coating J2 has displayed a more uniform and denser foam structure which can strongly adhere to the steel substrate after the fire test. This rigid formation can contribute to the enrichment of the fire-protective performance on the coated steel substrate and subsequently lead to better characteristic properties of the coating. Therefore, a fire protective performance which indicates the competency of the char layer in fire resistance is highly dependent on its physical structure of the coating [27]. Hence, when selecting the specified formulation of flame-retardant fillers, it is worth noting that the coating needs to be able to enhance the intumescent coating performance in terms of fire protection and thermal properties.

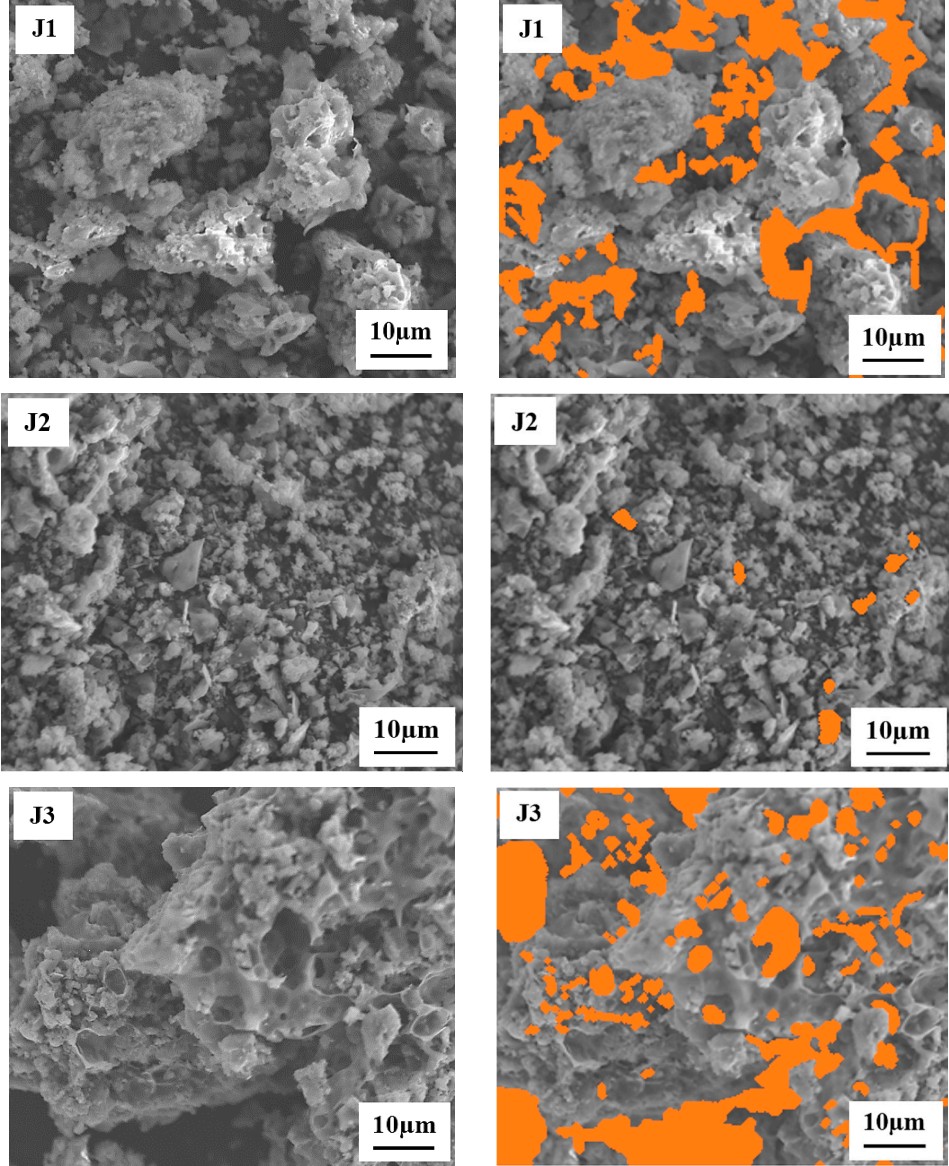

**Figure 7.** *Cont.*

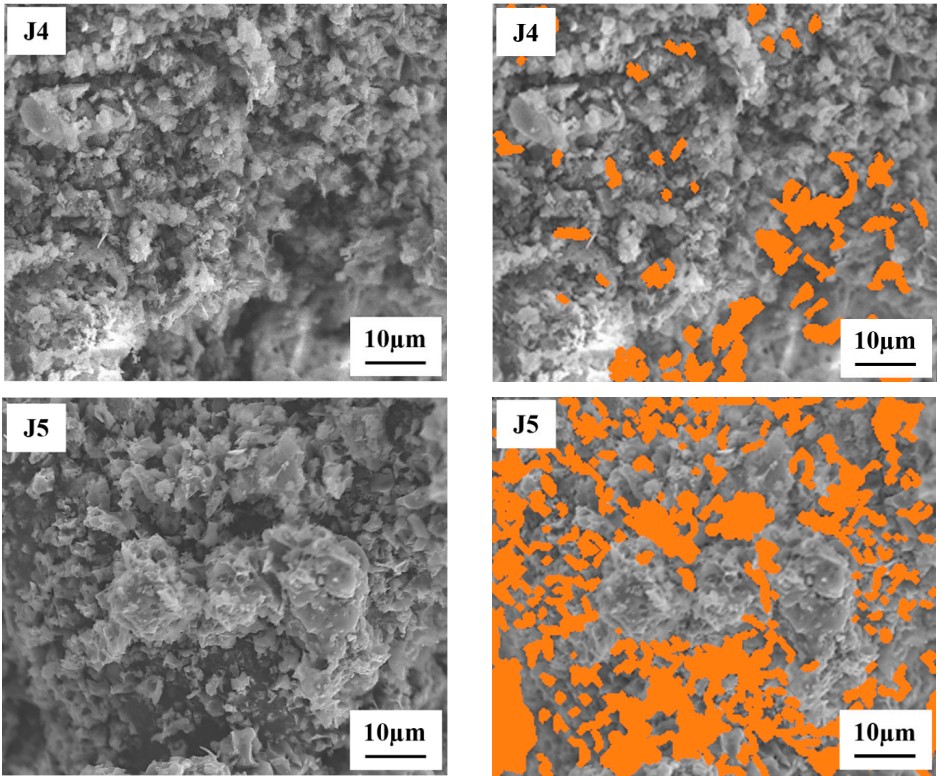

**Figure 7.** Surface morphology of the various char layer samples.

### 3.4. Adhesion Strength of Coated Samples

This test is carried out to determine the adhesion strength of the coated samples. The average adhesion strength amongst all coating samples is tabulated and presented in Figure 8. All the coating samples show good adhesion strength values. The coatings J1–J5 have showed higher adhesion strength values ranging from 1.05 to 1.45 MPa. The increased adhesion strength can be interpreted due to the influence of the carbonyl group between VA copolymer and the surface oxide or hydroxide of steel substrate that has created a stronger carbon–hydrogen (CH) bond in the mixture of the coating. The intermolecular bonding between the flame-retardant fillers' molecules has further improved the interaction between the coating and the steel substrate as well as between themselves [28]. This is not to disregard the carbon–hydrogen bonding, which in fact has a significant role in the adhesive properties of the coating. The carbon–hydrogen bonding is formed between a strongly electronegative atom that is usually oxygen and a hydrogen atom. The hydrogen bonds are formed between a monomer of the VA copolymer molecule and the steel substrate.

Judging from the results obtained, it is revealed that the coating J5 with the addition of $Mg(OH)_2$ and $CaCO_3$ have shown the highest adhesion strength that is 1.45 MPa. The increase in adhesion strength of coating J5 is due to the strong bonding strength between the steel substrate surface and the VA copolymer binder with the addition of $Mg(OH)_2$ and $CaCO_3$ fillers, which distribute stresses effectively. Mechanical stress can encourage the development of voids in the matrix caused by the loss of adhesion between the coating and the filler [29], whereas the coatings J2 and J4 with the addition of $Al(OH)_3$ + CES and $Mg(OH)_2$ + CES show higher adhesion strength of 1.25 MPa and 1.26 MPa, respectively, as compared to other coatings. The addition of CES filler has indeed shown improvement in adhesion strength due to the improved reinforcement properties between the CES filler and VA copolymer binder [30]. However, the adhesion strength of coating J4 with the addition of $Mg(OH)_2$ and CES is higher than the adhesion strength of the coating J2 with the addition of $Al(OH)_3$ and CES due to the $Al(OH)_3$ filler leading to the reduction of adhesion strength. This is due to the aluminium hydroxide which reduces the adhesion strength in the particular case under the influence of moisture where the

oxide layer transforms into a hydroxide with a morphological change. The adhesion strengths of the coatings J1 and J3 with the addition of calcium silicate have showed lower adhesion strength of 1.05 and 0.95 MPa, respectively. This indicates that the coating formulation incorporated with calcium silicate would decrease the adhesion strength. Therefore, choosing an appropriate flame-retardant filler strongly influences the interfacial bonding strength between the intumescent coating and the steel substrate [31].

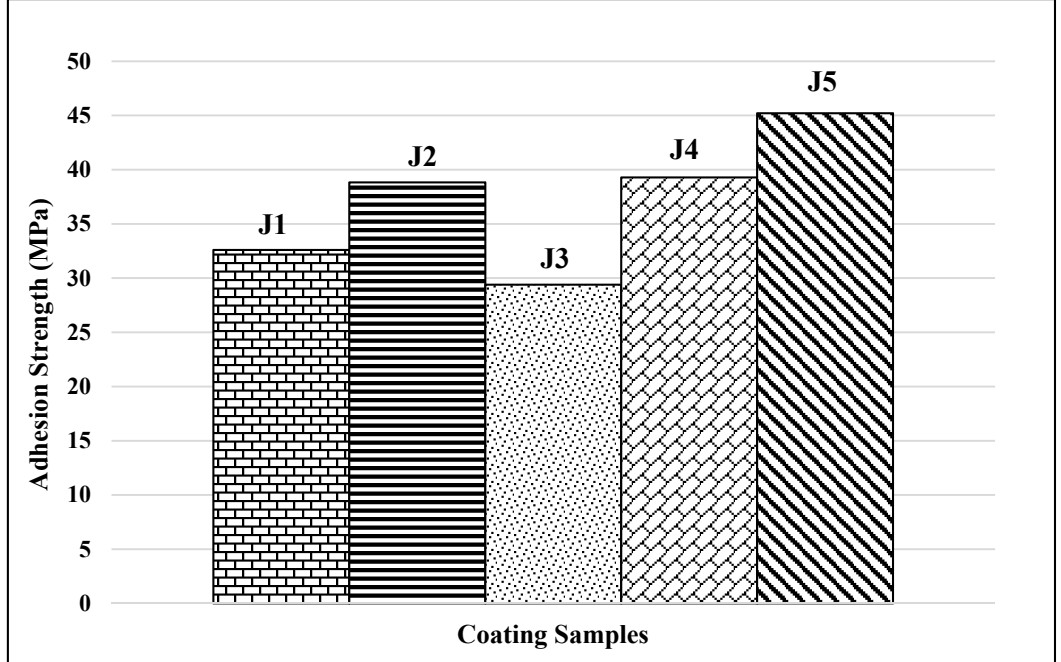

**Figure 8.** Adhesion strength of the coating samples.

### 3.5. Freeze-Thaw Cycle Test

This test is conducted to evaluate the effect of room temperature, freezing temperature and high temperature, symbolizing different weather conditions such as winter, spring, autumn, and summer on the coating surface. Figure 9 shows the results of all coated samples before and after 21 freeze–thaw cycle tests. The coated samples are visually inspected for cracks, blisters, coagulated particles, and changes of color. In this research study, all the coated samples remain unchanged and free from any visible cracks and coagulated particles after freeze–thaw cycles. All coating samples have proven to be efficient in withstanding the highly destructive forces of cyclic freezing and thawing.

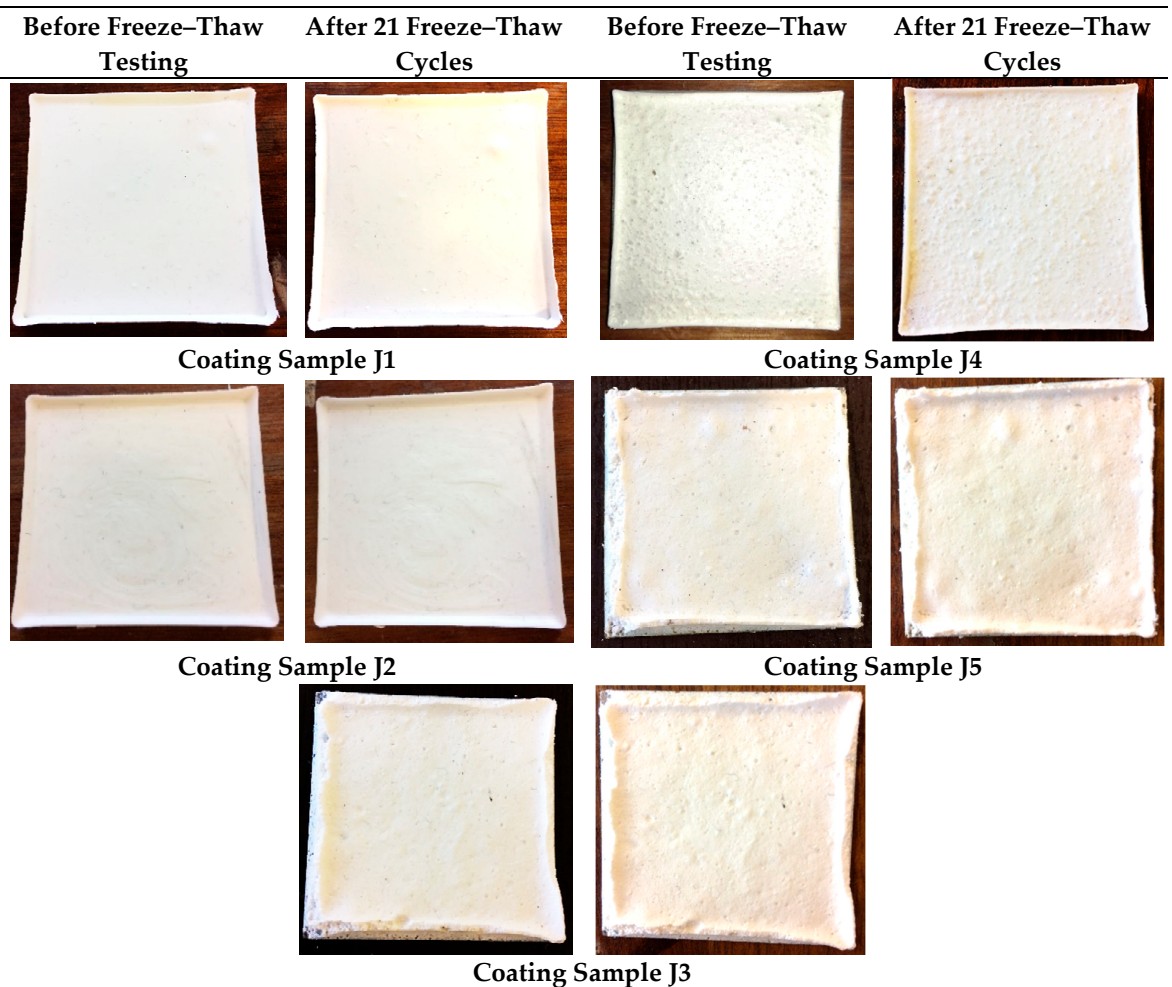

| Before Freeze–Thaw Testing | After 21 Freeze–Thaw Cycles | Before Freeze–Thaw Testing | After 21 Freeze–Thaw Cycles |

**Coating Sample J1**　　　　　　**Coating Sample J4**

**Coating Sample J2**　　　　　　**Coating Sample J5**

**Coating Sample J3**

**Figure 9.** Before and after the freeze–thaw cycles test of the coated samples.

### 3.6. Thermogravimetric Analysis (TGA) Test

This test is to determine the thermal degradation by calculating the residual weight of the intumescent coatings. From the results obtained in Figure 10 under nitrogen gas, the thermal decomposition is mainly observed at a temperature between 200–450 °C. However, the thermal decomposition can also occur at different stages. In this study, the general thermogram profiles are quite similar but the curves are found to be slightly shifted towards higher temperature ranges with gradual increments in heating rates. From 30 to 100 °C, around 2% of the initial weight is lost. This is possibly due to the release of moisture. Between 100–200 °C, J1 and J2 showed slight weight loss of approximately 1%–2%, where J3, J4 and J5 showered around 5% of the initial weight losses. Between 200–300 °C, a weight loss of 10% of the initial weight is observed. Between 300–450 °C, a measured weight loss of approximately 45% is calculated. However, between 450–1000 °C, the weight loss is only 10% of the initial weight observed. This can be attributed to the fact that the degradation/decomposition reaction has reached a relatively stable stage in the mixture.

In summary, the residual weights of J1, J2, J3, J4, and J5 at 950 °C are 34.28%, 34.55%, 32.77%, 33.82%, and 31.34% respectively. The TG curves have clearly demonstrated that the residue weight of coating sample J2 has the highest residual weight which indicates that the addition of aluminum hydroxide and CES fillers content in intumescent binder could enhance anti-oxidation of the coatings [32].

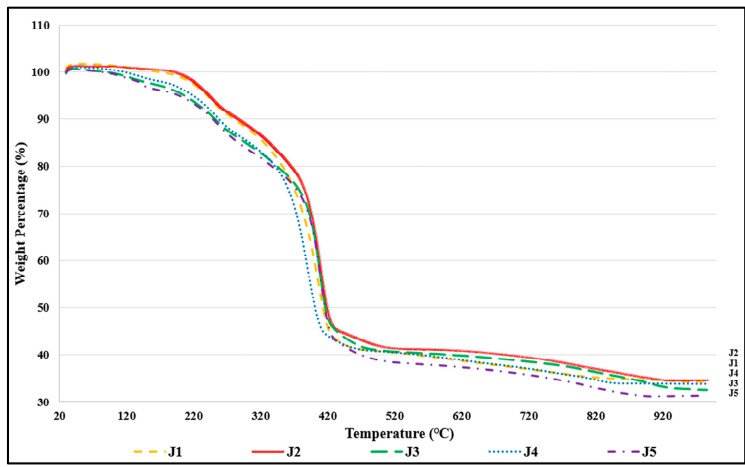

**Figure 10.** TGA curves of each coating sample.

*3.7. Testing for Fire-Resistant Timber Door Prototypes*

3.7.1. Small Scale Fire Test

This test is to characterize the fire protective performance of the fire-resistant timber door prototypes (P1–P5). The results of each coating sample plotted as a function of time are presented in Figure 11. The results show that the temperature of each fire-resistant timber door prototype has increased gradually. From the results obtained, all the temperatures of prototypes increase significantly in the first 15 min, revealing a similar pattern. This phenomenon is because of the good thermal insulation of vermiculite and perlite. These porous materials can dissipate the heat when exposed to fire. After 15 min, P4 has started to increase rapidly and gradually until 20 min. This may be due to the physical and chemical reactions of the intumescent coating during the fire test. After 20 min, all the prototypes have showed a similar pattern of temperature profiles until reaching 120 min. On top of that, P2 has the lowest temperature of 140.7 °C at the center of the prototype as compared to other types of prototypes. It has also been indicated that this prototype, P2, has resulted in the best fire protective performance in retarding the fire.

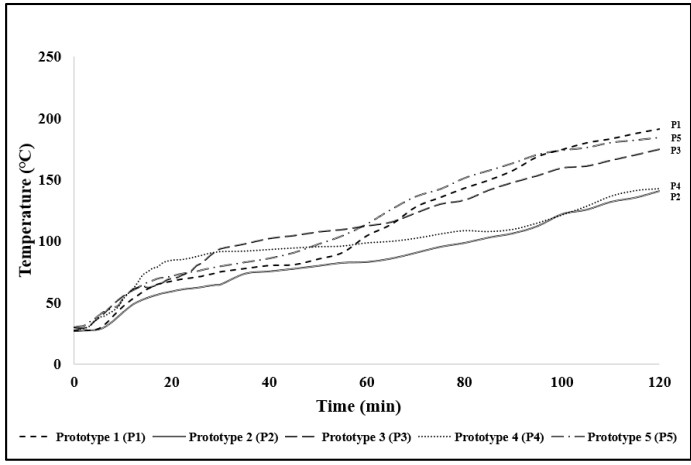

**Figure 11.** Evolution of temperature of fire-resistant timber door prototypes.

3.7.2. Fire Endurance and Temperature Rise Tests

As a building safety requirement, the safety and the reliability of the fire door is the key aspect to determining the effectiveness during the outbreak of the fire. In this research project, there are only commercial prototypes, and the lowest equilibrium temperature of P2 have been selected to undergo

the fire endurance [33] and temperature rise tests [34] to compare the fire-resistance rating and heat transmission rate by visual observation/inspection. The fire endurance test is to characterize the ability of the fire-resistant timber door prototypes to survive at a specified fire condition for a certain time without an integrity failure or a significant leakage. In this project, the magnesium oxide board acts as the commercial fire-resistant door prototype, which is used to compare with prototype P2. The results of fire endurance and temperature rise of prototypes after the 2 h fire is shown in Figure 12. As a result, both prototypes achieved the 2-h fire-resistance rating without an integrity failure or a significant leakage.

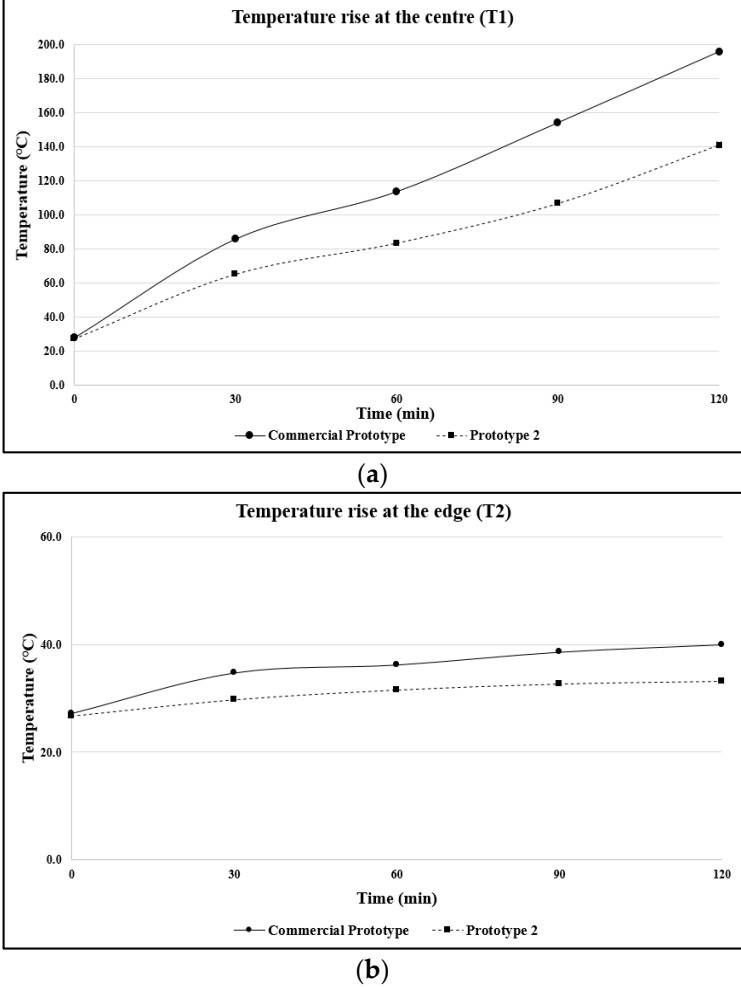

(**a**)

(**b**)

**Figure 12.** Temperature rise ratings at the center (T1) (**a**) and the edge (T2) (**b**) of the P2 and commercial prototypes.

Based on the results obtained, P2 has shown the best relatively low-temperature rise rating during a 120 min fire, which can be considered to have possessed a good quality of low heat transmission at T1 and T2, as compared to a commercial prototype. By analyzing the temperature rises at different points (T1 and T2), P2 has also indicated the best fire retardancy properties by promoting a lower equilibrium temperature than the commercial prototype. Moreover, the maximum temperatures of the commercial prototype are merely 195.6 (T1) and 40.0 °C (T2), respectively. For P2, the maximum temperatures at T1 and T2 are merely 140.7 and 32.0 °C, respectively. This has clearly indicated that P2 has a higher fire resistance than the commercial prototype. This phenomenon can be due to the density differences, i.e., the density of P2 (density = 636.45 kg/m$^3$) is lower as compared to the current existing commercial prototype (density = 873.71 kg/m$^3$).

Besides that, an appropriate combination of intumescent flame-retardant coating with vermiculite and perlite can also lead to a lower heat transmission and reveal less surface cracks, resulting in the reduction of temperature up to 54.9 °C as compared to the commercial prototype after the 2 h fire test, as seen in Figure 13. Furthermore, the higher temperatures of the commercial prototype measured at T1 and T2 are also higher than those measured for the P2 prototype as evident in Figure 13b with numerous cracks on the surface of the prototype providing higher rates of heat transfer to the backside of the prototype.

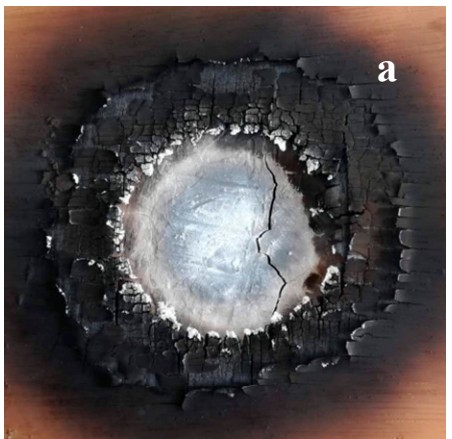 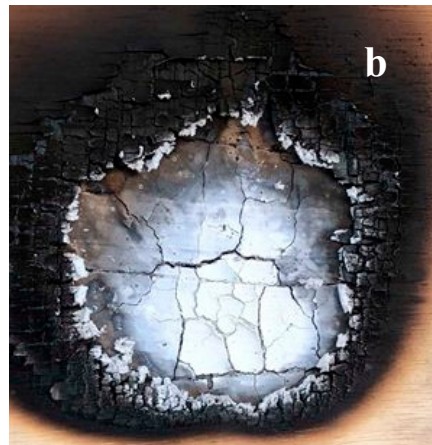

**Figure 13.** Fire door prototypes (**a**) P2 and (**b**) commercial after the 2-h fire test.

## 4. Conclusions

From the series of experimental tests, the following conclusions can be deduced. The selection of suitable combinations of flame-retardant materials can directly affect the fire-protective performance and the mechanical properties of the intumescent flame-retardant coating as well as the fire door prototype. Amongst all the coating samples used/manufactured in this research project, the coating J2 with the addition of 3 wt.% of aluminum hydroxide and 3 wt.% of renewable CES bio filler is concluded to have the best fire-protective performance, thermal, mechanical, physical, and chemical properties. On top of that, the P2 prototype with the addition of J2 intumescent coating reveals to a better fire protective result (temperature reduction by up to 54.9 °C), as compared to the existing commercial fire-resistant timber door prototype.

To summarize the conclusions based on the experimental results, an innovative intumescent flame-retardant paint incorporated into the lightweight fire-resistant timber door prototype (P2) has indeed demonstrated to be more efficient in reducing the heat transmission by maintaining its integrity without showing a significant leakage against the 2-h fire test.

**Author Contributions:** Conceptualization, J.J.K.Y and M.C.Y.; Methodology, J.J.K.Y.; Software, J.J.K.Y.; Validation, M.C.Y., M.K.Y. and L.H.S.; Formal Analysis, J.J.K.Y. and M.C.Y.; Investigation, J.J.K.Y. and M.C.Y.; Resources, M.C.Y.; Data Curation, J.J.K.Y. and M.C.Y.; Writing–Original Draft Preparation, J.J.K.Y.; Writing–Review & Editing, M.C.Y. and M.K.Y.; Visualization, M.K.Y. and L.H.S.; Supervision, M.C.Y.; Project Administration, M.C.Y. and M.K.Y.; Funding Acquisition, M.C.Y.

**Funding:** This project was funded by University of Tunku Abdul Rahman Research Scheme (UTARRF) with project number: IPSR/RMC/UTARRF/2018-C2/Y03.

**Acknowledgments:** The authors would like to thank University of Tunku Abdul Rahman (UTAR) for the UTARRF funding support.

**Conflicts of Interest:** The authors declare no conflict of interest.

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
