# Peer review of "Preparation of Intumescent Fire Protective Coating for Fire Rated Timber Door"

_coatings, doi:10.3390/coatings9110738_

Round 1
Reviewer 1 Report
The paper addresses the fire-resistance, physical, chemical, and mechanical properties of the Intumescent flame-retardant coating using several experimental tests. Moreover, the authors show experimental results on two Fire Rated timber door. In general, the paper content is relevant from the scientific point of view and addresses important issues relevant for the fire safety engineering community and building construction sector.
The paper is recommended for publication after some revisions and suggestions.
1- Page 1 line 36-39. The meaning of the sentence is unclear.
2- Page 1 line 43-44. “insulating the heat.”. unclear
3- Page 2 line 45. “inboard.”. unclear
4- Page 3 line 98. “experimental program” instead of “experimental”
5- Did you follow any regulation for the experimental tests?
6- How did you apply the intumescent coating? Did you check the thickness homogeneity on the sample?
7- Bunsen burner test: “Which is the distance between the Bunsen burner flam and the protected steel plate?”
8- Furnace test:
-which is the input fire curve? You indicate just the maximum temperatures.
- which is the furnace dimension? Is it electrical or not?
- did you measure the thickness variation during the tests? See “de Silva et al., 2019.”
9- Adhesion strength test: why did you consider 1 mm of intumescent coating and not 2 mm as the previous tests?
Can the thickness of the coating influence the adhesion strength? You should consult “Bilotta et al., 2016.”
10- Page 5 line 151: F= crack charge. Unclear
11- Page 6 line 159: Please indicate the reference.
12- Testing for fire resistant timber door prototype: How did check the fire insulation and integrity of the timber door? Please insert a detailed photo/image of the prototype. Why did you use the Bunsen burner and not the furnace? Generally, the fire resistance of door is assessed with furnace tests.
13- Page 16 line 163-167. The meaning of the sentences is unclear. Check the English please.
14- What do you mean by “fire endurance”?
15- Page 7 line 177: “…door if it could 178 maintain its integrity when exposed to 2-hour fire (at about 1000 °C) without an integrity failure or displaying a significant leakage.” Please insert the reference.
16- In all the tests…did you test the bare steel?
17- Please clarify the definition and the meaning of the equilibrium temperature.
18- Page 8 lines 202-203 please check the English.
19- Page 9 lines 225-226 please check the English. “to failing???”
20- Page 9 lines 226-229 please check the English. “increased???”
21- Figure 6. Insert “J1, J2…” on each column of the histogram…the meaning could be more clear.
22- Page 10 line 266. What do you mean by “segregate”?
23- Page 15 line 342: “How did you measure and check the integrity?”
24- Figure 12- Please insert in the graphs the input temperature.
25- Please clarify the meaning of fire resistance.
Author Response
Response to reviewers
The authors thank the reviewers for their time in reviewing the manuscript. Their kind remarks provide valuable insight, stimulate profound discussions, point out errors and ambiguity in our writing, and greatly improve our work. We hereby submit the following responses, along with a revised version of the manuscript, in which all changes made are MARKED in YELLOW colour. We hope that all the issues raised by the reviewers have been addressed.
Response to Reviewer #1
Comment by reviewer #1: The paper addresses the fire-resistance, physical, chemical, and mechanical properties of the Intumescent flame-retardant coating using several experimental tests. Moreover, the authors show experimental results on two Fire Rated timber doors. In general, the paper content is relevant from the scientific point of view and addresses important issues relevant for the fire safety engineering community and building construction sector.
Point 1: Page 1 line 36-39. The meaning of the sentence is unclear.
Response 1: As known, the fire-resistant door is commonly made of timber wood, gypsum board, magnesium oxide board and so on, but none of the commercialised fire-resistant door is made of intumescent binder (which is made of intumescent flame-retardant coating – IFRC). This sentence is emphasized that the novelty of our research is fabricating a new and lightweight fire-resistant door using intumescent binder that can drastically improve the present technical performance of fire resistance aspect of all commercially available fire-resistant doors.
Point 2: Page 1 line 43-44. “insulating the heat.”. unclear
Response 2: “Insulating” has been revised to “isolating” the heat means blocking and/or resisting the heat produced by the fire during any outbreak of fire. When there is a fire outbreak, the temperature of heat produced by the fire can be extremely high and can burn those victims to death who are trapped inside an enclosure and/or within the building to be very uncomfortable and it can lead to suffocation and eventually death.
Point 3: Page 2 line 45. “inboard.”. unclear
Response 3: Intumescent flame-retardant coating (IFRC) can be coated as a thin film (eg. Paint) or acted as a binder (eg: fire resistant door) of PFPs to enhance the fire resistance performance.
Point 4: Page 3 line 98. “experimental program” instead of “experimental”
Response 4: Thanks for the correction. The “experimental” has been revised to “experimental program” in page 3 line 98.
Point 5: Did you follow any regulation for the experimental tests?
Response 5: The experimental works are mainly designed for laboratory tests and partial standard test (e.g. adhesion strength test is partially in accordance with ASTM D41) in this project, which are cited from other published articles. Thus, we have not complied with full standard tests.
Point 6: How did you apply the intumescent coating? Did you check the thickness homogeneity on the sample?
Response 6: We have applied the intumescent paint using a gun sprayer on steel plate substrates layer by layer until the certain thickness stated in Section 2.2. Testing for coated samples for testing purpose. Yes, indeed. We have also checked the thickness homogeneity of each coating sample before undergoing a series of planned tests.
Point 7: Bunsen burner test: “Which is the distance between the Bunsen burner flame and the protected steel plate?”
Response 7: The distance between the Bunsen burner flame and the protected steel plate substrate is 7 cm (i.e., from the flame nozzle to the surface of the coating on the steel plate) for the ease of comparison towards all the samples. The distance is according to the flame size that contacts to the surface of the coating as depicted in the photographic view below.
Point 8: Furnace test:
-which is the input fire curve? You indicate just the maximum temperatures.
- which is the furnace dimension? Is it electrical or not?
- did you measure the thickness variation during the tests? See “de Silva et al., 2019.”
Response 8: The model of the furnace has been included in page 5 line 143 and the heating rate in the research has been added in line 144. During the duration of the experimental tests, we have not measured the input fire curve and thickness variation of the char layer. We have only measured the thickness of the coating (before fire test) and char layer (after fire test). Thanks for the suggestion, we will include to the future research program.
Point 9: Adhesion strength test: why did you consider 1 mm of intumescent coating and not 2 mm as the previous tests?
Can the thickness of the coating influence the adhesion strength? You should consult “Bilotta et al., 2016.”
Response 9: Our apology for the wrong description of the thickness of the intumescent coating used. The “1 ±0.1 mm” has been altered to “2 ±0.2 mm” in page 5 line 144 and 146. With reference to Bilotta et al., 2016, it is indeed indicated that the greater the thickness of the coating will give the lower the adhesion force measured. However, for the ease of comparison throughout all the tests with the coating samples, we have used 2 ±0.2 mm as the thickness of the coating to do the test. Although the results may not as good as the coating with the thickness of 1 ±0.1 mm, it can still achieve a good result in mechanical properties.
Point 10: Page 5 line 151: F= crack charge. Unclear
Response 10: F is referred to the maximum load at rupture. The paper has been revised accordingly in pages 5 line 151.
Point 11: Page 6 line 159: Please indicate the reference.
Response 11: Thanks for the advice. The reference has been added and updated in page 6 line 167 accordingly.
Point 12: Testing for fire resistant timber door prototype: How did check the fire insulation and integrity of the timber door? Please insert a detailed photo/image of the prototype. Why did you use the Bunsen burner and not the furnace? Generally, the fire resistance of door is assessed with furnace tests.
Response 12: In this paper, the fire resistant timber door prototypes are fabricated with intumescent binder and plywood. After that, a digital handheld thermometer is placed at the backside of the prototype to measure the temperature profile throughout the 2-hour testing. Below is the schematic of the experimental setup for fire resistant timber door prototype and a detailed photo of the prototype.
The integrity of the timber door prototype is determined by observing the width of crack line or hole on the surface of the tested timber prototype for 2 hours during the Bunsen burner test. In this project, Bunsen burner test is carried out to determine the fire endurance test instead of using furnace due to the flame (about 1000°C) from Bunsen burner is closer to the real-life fire. As for the large-scale fire door test, it will be conducted at SIRIM laboratory in the next paper. Thanks for the suggestion.
Point 13: Page 16 line 163-167. The meaning of the sentences is unclear. Check the English please.
Response 13: Thanks for your advice. The paper has been revised in page 6 line 170 – 171.
Point 14: What do you mean by “fire endurance”?
Response 14: Fire endurance is referred to the ability to survive a specified fire condition for a certain time without any integrity failure or a significant leakage. In the paper, we have intended to compare the intumescent binder fire-resistant prototypes via defined the fire endurance whether it can withstand the 2-hour fire proofing with high temperatures.
Point 15: Page 7 line 177: “…door if it could 178 maintain its integrity when exposed to 2-hour fire (at about 1000 °C) without an integrity failure or displaying a significant leakage.” Please insert the reference.
Response 15: Thanks for the advice. The reference has been added in page 7 line 182.
Point 16: In all the tests…did you test the bare steel?
Response 16: The bare steel has indeed been tested by (Yew and Ramli Sulong, 2012), Fire-resistive performance of intumescent coating for steel paper. Authors did not include the test for bare steel in this experimental work.
Point 17: Please clarify the definition and the meaning of the equilibrium temperature.
Response 17: In this paper, equilibrium temperature is referred to a state in which the temperature reached at the steady state.
Point 18: Page 8 lines 202-203 please check the English.
Response 18: Thanks for advice, we have rectified it accordingly.
Point 19: Page 9 lines 225-226 please check the English. “to failing???”
Response 19: Thanks for advice, we have corrected it accordingly.
Point 20: Page 9 lines 226-229 please check the English. “increased???”
Response 20: Thanks for advice, we have rectified it accordingly.
Point 21: Figure 6. Insert “J1, J2…” on each column of the histogram…the meaning could be more clear.
Response 21: Thanks for your advice. Figure 6 has been revised.
Point 22: Page 10 line 266. What do you mean by “segregate”?
Response 22: The meaning of segregate in page 10 line 266 means isolate. When the char layer is expanded and formed a uniform and denser foam structure during fire outbreak, the char layer will eventually block the flame and heat to penetrate. Therefore, it can gain more time for the victims to escape and the buildings to collapse.
Point 23: Page 15 line 342: “How did you measure and check the integrity?”
Response 23: Thanks for your advice. As explained in Response 12, the integrity of the timber door prototype is determined by observing the width of crack line or hole on the surface of the tested timber prototype for 2 hours during the Bunsen burner test.
Point 24: Figure 12- Please insert in the graphs the input temperature.
Response 24: Thanks for your advice. The graph in Figure 12 has been revised.
Point 25: Please clarify the meaning of fire resistance.
Response 25: Fire resistance is a measurement to determine the ability of an element structure to withstand the effects of fire without impairing its properties. Fire resistance is also known as the property of materials that retards or prevents the passage of excessive heat, hot gases or flames under conditions of use.

Reviewer 2 Report
This paper studied a series of comprehensive experimental work on Intumescent coating on the performance of fire protective coatings. This work introduces 5 different formulations with different flame-retardant fillers and a series of tests to demonstrate their fire protective performance. Those tests which include the Bunsen burner test, Furnace test, Surface morphology using SEM, Adhesion strength test, Freeze-thaw cycle test, and fire tests, provides a comprehensive understanding and full evaluations of the performance of those 5 samples. The experimental procedures are well explained and results are clear to support the conclusion statements made in this paper. I would recommend this paper to be published in this journal. If condition provided, additional full-scale fire tests are recommended for better evidence.
Author Response
Response to Reviewer #2 Comments:
Comment by Reviewer #2: This paper studied a series of comprehensive experimental work on Intumescent coating on the performance of fire protective coatings. This work introduces 5 different formulations with different flame-retardant fillers and a series of tests to demonstrate their fire protective performance. Those tests which include the Bunsen burner test, Furnace test, Surface morphology using SEM, Adhesion strength test, Freeze-thaw cycle test, and fire tests, provides a comprehensive understanding and full evaluations of the performance of those 5 samples. The experimental procedures are well explained and results are clear to support the conclusion statements made in this paper.
Point 1: I would recommend this paper to be published in this journal. If condition provided, additional full-scale fire tests are recommended for better evidence.
Response 1: Thanks for the Reviewer’s affirmation of our manuscript. Your encouragement has inspired us. Authors will conduct full-scale tests soon as recommended for better evidence and fulfil the standard time-temperature fire curve.

Reviewer 3 Report
The paper presents a preparation of intumescent fire protective coating for fire rated timber door. The topic is contemporary and interesting. However, in my opinion, some major corrections and supplement analytical methods should be provided in the manuscript to fulfil the conditions needed for its publication in Coatings.
General comment:
Cone calorimetric and thermogravimetric analysis of the prototypes should be done since they are crucial for the characterisation of FR material.
Title: The words ‘’Advances in’’ should be replaced with more appropriate once, for example, Preparation of
Abstract: The experimental part and the results should be rewritten to be more informative. Furthermore, there is no information about the use of vermiculite and perlite as FR filler in the experimental part. Authors should verify this statement in the abstract.
Experimental: Line 104: The abbreviation of the chicken eggshell powder should be included since it is used in Table 1.
Table 1 should be rewritten since it structure is not appropriate. Namely, there are two tables in one structure, which is not acceptable because the table header rows do not identify the content of all columns. In the first column, VA should be used instead of polymer binder.
Figure 1 to Figure 4 should be deleted since they do not give additional information about the experiment.
Figure 8 should be corrected: delete the title and the legend, introduce the codes of the coating samples on x axis.
Figure 9 should be deleted since it does not present the results of the research but already known findings. Otherwise, the hydrogen bonding should be confirmed with an appropriate analytical method.
Author Response
1. Response to Reviewer #3 Comments:
Comment by Reviewer #3: The paper presents a preparation of intumescent fire protective coating for fire rated timber door. The topic is contemporary and interesting. However, in my opinion, some major corrections and supplement analytical methods should be provided in the manuscript to fulfil the conditions needed for its publication in Coatings.
Point 1: Cone calorimetric and thermogravimetric analysis of the prototypes should be done since they are crucial for the characterisation of FR material.
Response 1: Thanks for the suggestion. However, it is not possible to carried out cone calorimetric in Malaysia, as there is no equipment in Malaysia. For the thermogravimetric analysis, we have conducted, and it will include in the next paper.
Point 2: Title:
The words ‘’Advances in’’ should be replaced with more appropriate once, for example, “Preparation of”
Response 2: The title has been revised from “Advances in” to “Preparation of”
Point 3: Abstract:
The experimental part and the results should be rewritten to be more informative. Furthermore, there is no information about the use of vermiculite and perlite as FR filler in the experimental part. Authors should verify this statement in the abstract.
Response 3: Thanks for your advice. We have included the use of vermiculite and perlite as FR filler in abstract (line 16 – 17) and explained about the vermiculite and perlite in the introduction (line 55 onwards). However, due to the proprietary ingredient of vermiculite and perlite at the current state, so we will not be revealed it now. Thanks for your understanding.
Point 4: Experimental:
Line 104: The abbreviation of the chicken eggshell powder should be included since it is used in Table 1.
Response 4: Thanks for your advice. The paper has been revised in page 3 line 108.
Point 5: Table 1 should be rewritten since it structure is not appropriate. Namely, there are two tables in one structure, which is not acceptable because the table header rows do not identify the content of all columns. In the first column, VA should be used instead of polymer binder.
Response 5: Thanks for your suggestion. Two tables have been separated to Table 1 and Table 2. The polymer binder has been revised in the first column of Table 1. The paper has been revised.
Point 6: Figure 1 to Figure 4 should be deleted since they do not give additional information about the experiment.
Response 6: Thanks for your advice. However, the reason of showing Figures 1-4 are because it is presentable, and readers will easily to understand the flow of the article with additional information about the project. Other reviewers have recommended to include the Figures. Thanks for your understanding.
Point 7: Figure 8 should be corrected: delete the title and the legend, introduce the codes of the coating samples on x axis.
Response 7: Thanks for the suggestion. The graph has been revised in Figure 8.
Point 8: Figure 9 should be deleted since it does not present the results of the research but already known findings. Otherwise, the hydrogen bonding should be confirmed with an appropriate analytical method.
Response 8: Thanks for your advice. Figure 9 has been deleted. The paper has been revised.

Round 2
Reviewer 3 Report
Regarding the authors answer: For the thermogravimetric analysis, we have conducted, and it will include in the next paper.
In my opinion, the results of the thermogravimetric analysis should be included into this paper, because these results give crucial information about the thermal behavior of the coatings.
Author Response
In order to accept your paper for publication, the Editors ask you to complete your results by adding thermogravimetric experiments.
To editor,
Thanks for the suggestion. In this paper, we have covered the title: "Preparation of Intumescent Fire Protective Coating for Fire Rated Timber Door" by using Bunsen burner, furnace, Scanning Electron Microscope, freeze-thaw stability test, and Instron Micro Tester to analyse and evaluate the performance of the sample coatings and prototypes towards fire resistance, freeze-thaw cycle, surface morphology, adhesion strength and fire endurance.
We would be happy if the TGA experiment can be excluded. Thanks for your consideration.
Regards,
Yew